# Evolving and Regularizing Meta-Environment Learner for Fine-Grained Few-Shot Class-Incremental Learning

**Li-Jun Zhao[1], Zhen-Duo Chen[1,*], Yongxin Wang[2], Xin Luo[1], Xin-Shun Xu[1]**
[1]School of Software, Shandong University, China
[2]School of Computer and Artificial Intelligence, Shandong Jianzhu University, China
`lj_zhao1028@163.com, chenzd.sdu@gmail.com, yxinwang@hotmail.com`
`luoxin.lxin@gmail.com, xuxinshun@sdu.edu.cn`

## Abstract

Recently proposed Fine-Grained Few-Shot Class-Incremental Learning (FG-FSCIL) offers a practical and efficient solution for enabling models to incrementally learn new fine-grained categories under limited data conditions. However, existing methods still settle for the fine-grained feature extraction capabilities learned from the base classes. Unlike conventional datasets, fine-grained categories exhibit subtle inter-class variations, naturally fostering latent synergy among sub-categories. Meanwhile, the incremental learning framework offers an opportunity to progressively strengthen this synergy by incorporating new sub-category data over time. Motivated by this, we theoretically formulate the FSCIL problem and derive a generalization error bound within a shared fine-grained meta-category environment. Guided by our theoretical insights, we design a novel Meta-Environment Learner (MEL) for FG-FSCIL, which evolves fine-grained feature extraction to enhance meta-environment understanding and simultaneously regularizes hypothesis space complexity. Extensive experiments demonstrate that our method consistently and significantly outperforms existing approaches.

## 1 Introduction

Fine-grained classification [12, 33] focuses on distinguishing instances from subordinate categories within the same meta-category (e.g., bird species, car models), and has been widely applied in domains such as ecological protection and security monitoring [28]. Traditionally, research has centered on improving fine-grained feature extraction using fixed and fully annotated datasets. However, in practical scenarios, the collection and annotation of fine-grained data are often limited by domain expertise and time costs, resulting in data scarcity and continuously evolving category sets. To address these challenges, recent studies have explored fine-grained classification under Few-Shot Class-Incremental Learning (FSCIL) [1, 24] setting. FSCIL enables models to learn new classes with limited samples after training on base classes, while retaining the ability to recognize all seen classes. Fine-Grained FSCIL (FG-FSCIL) [18] thus offers a practical and efficient solution for updating models with new fine-grained category information under limited data conditions.

Due to the high discriminative difficulty of fine-grained categories, FG-FSCIL research draws on traditional ideas from fine-grained classification methods and places greater emphasis on extracting fine-grained discriminative features. However, due to the limited training samples available during incremental sessions, this idea still settles for the fine-grained feature extraction capabilities learned

---

*Corresponding author

39th Conference on Neural Information Processing Systems (NeurIPS 2025).

from the base classes. Notably, fine-grained data differs from conventional data in that its sub-categories exhibit smaller inter-class variations, which naturally facilitates synergy across sub-categories. Meanwhile, the incremental learning setting provides the opportunity to progressively enhance this synergy by continuously incorporating new sub-category data over time. Motivated by data characteristics and the learning scenario, we aim to develop a model that not only refines its understanding of the meta-category environment using limited incremental data, but also leverages this evolving understanding to improve classification across all fine-grained sub-categories.

In this paper, we provide a theoretical formulation of the FSCIL problem and derive generalization error bounds for the expected error incurred when learning sub-category samples within a shared meta-environment. While traditional efforts to enhance fine-grained feature extraction help reduce empirical error and thus lower the bound, our analysis reveals that further reducing the bound relies on minimizing the complexity of both per-session sub-categories and the overall meta-environment. Crucially, we show that leveraging incremental sub-categories to enhance understanding of the meta-environment is a breakthrough point. The derived bounds also offer valuable guidance for achieving this under limited training data. Guided by these theoretical insights, we propose a novel Meta-Environment Learner (MEL) for FG-FSCIL, comprising two key processes: Evolving and Regularizing. Specifically, we introduce an incrementally optimized meta-category vector to enable evolving fine-grained feature extraction, enhancing the feature extractor's understanding of the meta-environment while avoiding unnecessary complexity. In parallel, sub-category relationship regularization guides feature space transformation and further constrains hypothesis space complexity, thus further regularizing two complexity terms. Our key contributions are summarized as follows:

- We theoretically formulate the FSCIL problem and analyze its generalization bound within the fine-grained meta-category environment, which is consistent with the core intuitions and motivations.

- Based on the derived generalization bound, we design a novel Meta-Environment Learner (MEL) for FG-FSCIL, which simultaneously evolves fine-grained feature extraction and regularizes hypothesis space complexity.

- Extensive experiments show that our method significantly outperforms both recent FSCIL methods and FG-FSCIL methods, demonstrating the effectiveness of our theoretically grounded design.

## 2 Related Work

**Fine-grained classification** [36, 23, 4] focuses on distinguishing visually similar sub-categories within a general meta-category, such as different species of birds, different breeds of dogs, or different models of airplanes. Compared to generic classification tasks, it demands models to capture subtle and localized discriminative features, as inter-class differences are often minimal. As a result, fine-grained classification methods [39, 31, 27] place greater emphasis on learning feature representations than conventional classification approaches. Given the challenges of collecting and annotating fine-grained data in real-world scenarios, Few-Shot Fine-Grained Classification (FSFG) [29, 10, 13, 14, 34] has emerged to address the problem of recognizing novel fine-grained categories using only a few labeled examples. However, existing works overlook the inherent potential of fine-grained sub-categories to collaboratively and incrementally enrich shared meta-category representations, which could serve as a natural foundation for continual learning.

**Few-Shot Class-Incremental Learning** (FSCIL) [5, 7, 26, 35] is a learning paradigm where the model first learns base classes with sufficient labeled data, then incrementally learns novel classes from only a few labeled samples. To improve performance, most methods [32, 37, 17, 11] leverage pretrained parameters to establish a strong initialization for subsequent learning, or employ data augmentation strategies, including the generation of pseudo-classes to facilitate later sessions. Although [6] includes experiments on fine-grained datasets, it remains a FSCIL method without specific discussion or methodological design for the characteristics of fine-grained data. Compared to traditional datasets, fine-grained datasets exhibit subtler inter-class differences and offer fewer labeled samples in the base session, making it challenging for conventional FSCIL methods to perform effectively in such settings. Recent work [18] has started to explore Fine-Grained Few-Shot Class-Incremental Learning (FG-FSCIL). However, similar to general FSCIL methods, they overlook

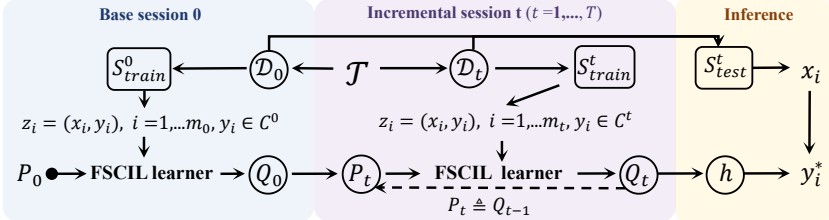

Figure 1: The FSCIL learner with environment $\mathcal{T}$, sample distributions $\mathcal{D}_t$, training set $S_{train}^t$, prior distribution $P_t$, posterior distribution $Q_t$, and a hypothesis $h$.

the inherent synergy among sub-categories in fine-grained data. As a result, the full potential of incremental learning is not fully exploited.

## 3 Methodology

In this section, we begin by providing a theoretical formulation of the FSCIL problem. Based on this, we derive a generalization bound, which forms the foundation for designing our FG-FSCIL method.

### 3.1 Problem Formulation

In FSCIL, the model is trained on a sequence of training sets $\{S_{train}^t\}_{t=0}^T$ from session $t$. Each $S_{train}^t$ comprises a set of pairs $z_i = (x_i, y_i)$, $i = 1, ..., m_t$, where $x_i$ is a sample from class $y_i \in \mathcal{C}^t$. For fine-grained dataset, all classes $\mathcal{C}^0 \cdots \cup \mathcal{C}^T$ belong to the same superclass. $\forall i, j$ and $i \neq j$, $\mathcal{C}^i \cap \mathcal{C}^j = \varnothing$. The training set $S_{train}^0$ in the base session contains sufficient samples, i.e., $m_0$ is much larger than others. In each subsequent incremental session ($t \geq 1$), the training set $S_{train}^t$ contains limited samples and follows an $N$-way $K$-shot format. This means there are only $K$ samples for each of the $N$ classes, i.e., $m_t = N \times K$ for $t \geq 1$. During the inference stage in session $t$, the test set $S_{test}^t$ is generated from all sample distributions $\mathcal{D}^0 \cup \mathcal{D}^1 \cdots \cup \mathcal{D}^t$ over all seen classes $\mathcal{C}^0 \cup \mathcal{C}^1 \cdots \cup \mathcal{C}^t$.

To model this setup, following [20, 2, 21], all classes from different sessions are assumed to share the same sample space $\mathcal{Z}$, hypothesis space $\mathcal{H}$, and loss function $\ell : \mathcal{H} \times \mathcal{Z} \to [0, 1]$. As shown in Figure 1, the FSCIL learner can access the dataset $S_{train}^t$ that is sampled i.i.d. from an unknown sample distribution $\mathcal{D}_t$ over the class set $\mathcal{C}^t$ in session $t$. All sample distributions $\mathcal{D}_t$ are drawn i.i.d from an unknown environment distribution $\mathcal{T}$ [3]. To solve the FSCIL problem, the learner takes as input a training set $S_{train}^t$ and a prior distribution $P_t$, and output a posterior distribution $Q_t = Q(S_{train}^t, P_t)$ over $\mathcal{H}$, that is, a mapping $\mathcal{Z} \times \mathcal{M} \to \mathcal{M}$, where $\mathcal{M}$ denotes the set of distributions over $\mathcal{H}$. The initial prior $P_0$ can be modeled as a Gaussian distribution over neural network weights [19]. For subsequent sessions, the prior $P_t$ is set to the posterior from the previous session, i.e. $P_t \triangleq Q_{t-1}$. Then, the expected error and empirical error for each sample distribution $D_t$ in session $t'$ are given by averaging over the posterior distribution $Q_{t'} \in \mathcal{M}$, namely $er(Q_{t'}, \mathcal{D}_t) \triangleq \mathbb{E}_{h \sim Q_{t'}} \mathbb{E}_{z \sim \mathcal{D}_t} \ell(h, z)$ and $\widehat{er}(Q_{t'}, S_{train}^t) \triangleq \mathbb{E}_{h \sim Q_{t'}} (1/m_t) \sum_{i=1}^{m_t} \ell(h, z_i)$ for a single hypothesis $h \in \mathcal{H}$.

### 3.2 Generalization Bounds for FSCIL and Meta-Category Environment

Based on the setting introduced in Section 3.1, we derive generalization bounds for FSCIL in session $t'$. Given the openness of the learning scenario, our goal is to enable the FSCIL learner to extract knowledge from the available categories that can benefit all classes within the same environment.

Therefore, the effectiveness of the posterior distribution $Q_{t'}$ is evaluated by the expected error incurred when learning classes drawn from the same environment $\mathcal{T}$, i.e., $er(Q_{t'}, \mathcal{T}) \triangleq \mathbb{E}_{\mathcal{D}_t \sim \mathcal{T}} \mathbb{E}_{h \sim Q_{t'}} \mathbb{E}_{z \sim \mathcal{D}_t} \ell(h, z)$. Specifically, we analyze the generalization error for both the newly introduced classes in each session and the overall environment. Formal proofs are provided in Section A.1 of the supplementary material.

**Theorem 3.1.** $\delta_t \triangleq \frac{\delta}{2(t'+1)\alpha_t}$ *For any $\delta \in (0, 1]$, the following bound holds uniformly for posterior distribution $Q_{t'}$, with probability at least $1 - \delta$ over a sequence of training set $S^0_{train}, ..., S^{t'}_{train}$,*

$$er(Q_{t'}, \mathcal{T}) \leq \sum_{t=0}^{t'} \alpha_t \widehat{er}(Q_{t'}, S^t_{train}) \tag{1}$$

$$+ \sum_{t=0}^{t'} \alpha_t \sqrt{\frac{1}{2(m_t - 1)} \left( \sum_{i=t}^{t'} D_{\mathrm{KL}}(Q_i \| P_i) + \log \frac{2(t'+1)\alpha_t m_t}{\delta} \right)} \tag{2}$$

$$+ \sqrt{\frac{1}{2t'} \left( D_{\mathrm{KL}}(Q_{t'} \| P_0) + \log \frac{2(t'+1)}{\delta} \right)}, \tag{3}$$

*where $D_{\mathrm{KL}}(\cdot \| \cdot)$ denotes the Kullback–Leibler (KL) divergence and $\alpha_t = |\mathcal{C}^t| / \sum_{i=0}^{t'} |\mathcal{C}^i|$.*

The overall bound comprises the **empirical error** (Equation 1) and two complexity terms: the **session complexity** from each sample distribution $\mathcal{D}_t$ (Equation 2) and the **environment complexity** across all sessions (Equation 3), which together offer potent insights for tackling the FSCIL challenge.

- **Insight 0.** *Reducing empirical error $\widehat{er}(Q_{t'}, S^t_{train})$ (Equation 1) can tighten the bound. Since $\alpha_0 > \alpha_t$ ($t \geq 1$), the error on $S^0_{train}$ plays a dominant role in this reduction.* In practice, focusing on fine-grained feature extraction on $S^t_{train}$ can effectively lower $\widehat{er}(Q_{t'}, S^t_{train})$.

Previous FG-FSCIL methods focus on training the feature extractor during session 0 and then freeze it, aligning with **Insight 0** to achieve a smaller overall empirical error. Other common techniques in FSCIL, such as initializing with pretrained models and applying data augmentation, essentially reduce the complexity terms—either by making the initial prior $P_0$ closer to the final posterior $Q_{t'}$, or by effectively increasing the number of training samples $m_t$.

A deeper understanding of Theorem 3.1 in the context of fine-grained data enables a more systematic analysis of the inherent synergistic learning potential that is often overlooked by existing FSCIL and FG-FSCIL methods. Based on widely recognized characteristics of fine-grained datasets, we begin by providing the following definition,

**Definition 3.2** (Meta-Environment)**.** *In fine-grained settings, where all sub-categories typically belong to the same meta-category, the shared environment $\mathcal{T}$—from which all sample distributions $\mathcal{D}_t$ are drawn—is referred to as the Meta-Environment. This structure implies smaller inter-class variation and facilitates positive transfer across sub-categories.*

Consequently, the FG-FSCIL learner is in a favorable position to leverage the information from seen sub-categories to extract transferable knowledge that benefits all sub-categories within the same meta-category. This makes the environment complexity term a breakthrough point for further reducing the overall generalization error in fine-grained settings:

- **Insight 1.** *As $t'$ increases, the environment complexity (Equation 3) tends to decrease, resulting in a tighter bound.* This suggests that leveraging sub-categories from incremental sessions to train the feature extractor can enhance its understanding of the overall meta-environment—consistent with the intuition and motivation discussed above.

- **Insight 2.** *Since $m_0 \gg m_t (t \geq 1)$, controlling $D_{\mathrm{KL}}(Q_t \| P_t)(t \geq 1)$ can effectively reduce the overall session complexity (Equation 2).* This indicates that while enhancing the feature extractor's understanding of the meta-environment, it is critical to limit significant changes in large parameter subsets during incremental learning.

- **Insight 3.** *Both complexity terms in Equations 2 and 3 characterize data-dependent hypothesis space complexity, which is often positively correlated with the number of learnable parameters.* From a generalization perspective of deep neural networks [38], hypothesis space complexity is primarily governed by classifier parameters, while feature space transformation may potentially simplify the classification.

### 3.3 Meta-Environment Learner for FSCIL

Building upon the theoretical insights above, we propose a novel Meta-Environment Learner (MEL) for FG-FSCIL. The overall architecture is illustrated in Figure 2. Unlike prior FG-FSCIL methods that

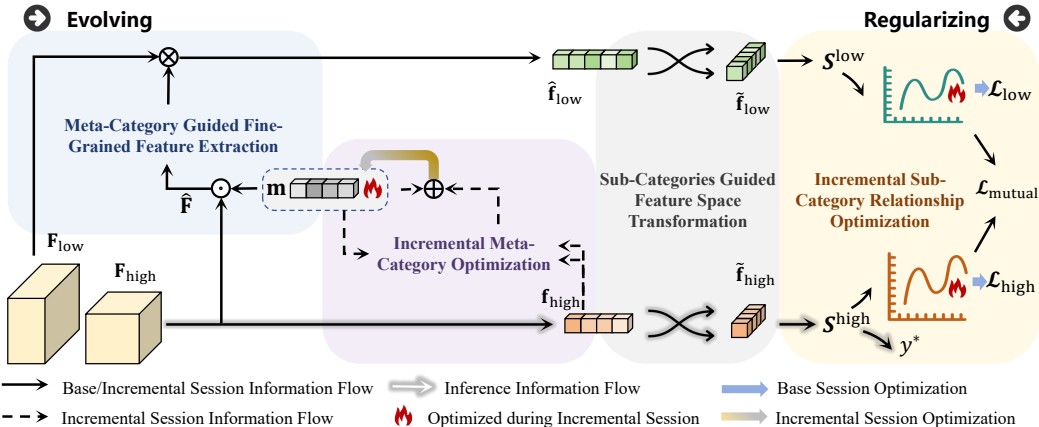

Figure 2: Overall method framework. During the incremental sessions, only Meta-Category Optimization and Sub-category Relationship Optimization are performed. Inference relies solely on high-level features.

freeze the feature extractor during incremental sessions, MEL continuously learns and leverages both the meta-category and its sub-categories. This allows the meta-environment to **evolve** progressively (Section 3.3.1) while remaining properly **regularized** (Section 3.3.2).

### 3.3.1   Evolving: Incremental Meta-Category Guided Fine-Grained Feature Extraction

Compared to conventional datasets, FG-FSCIL aims to minimize empirical risk (Equation 1) by identifying subtle differences among similar sub-categories through an evolving process (cf. **Insight 0**). To continuously optimize the fine-grained feature extraction process without significantly increasing the complexity terms (Equations 2 and 3), we avoid introducing additional heavy modules and instead primarily leverage the existing backbone structures (cf. **Insight 3**). Accordingly, for any backbone, we denote deeper-layer outputs as high-level semantic features $\mathbf{F}_{\text{high}} \in \mathbb{R}^{C_h \times H_h \times W_h}$, and shallower-layer outputs as low-level fine-grained features $\mathbf{F}_{\text{low}} \in \mathbb{R}^{C_l \times H_l \times W_l}$. Their average pooled representations are denoted as $\mathbf{f}_{\text{high}} \in \mathbb{R}^{C_h}$ and $\mathbf{f}_{\text{low}} \in \mathbb{R}^{C_l}$, respectively.

**Meta-Category Guided Fine-Grained Feature Extraction.**   With a lightweight design, our approach not only utilizes high-level semantic information but also integrates fine-grained details from low-level features. Specifically, to capture instance-specific cues, the high-level semantic features $\mathbf{F}_{\text{high}}$ of each instance can be utilized to spatially localize critical fine-grained features in $\mathbf{F}_{\text{low}}$. To further refine these cues across samples by leveraging the inherent correlations among fine-grained categories, we introduce a learnable meta-category vector $\mathbf{m} \in \mathbb{R}^{C_h}$, which encodes meta-category knowledge in a channel-wise manner with minimal parameter overhead (cf. **Insight 2**),

$$\hat{\mathbf{F}} = \mathbf{F}_{\text{high}} \odot \sigma(\mathbf{m}) \in \mathbb{R}^{C_l \times H_l \times W_l}, \tag{4}$$

where $\sigma$ denotes the leaky ReLU activation function and $\odot$ indicates element-wise multiplication. To align with the spatial resolution of the low-level feature map $\mathbf{F}_{\text{low}}$, we upsample the cues $\hat{\mathbf{F}}$, and then reduce the channel dimension to generate $N$ attention masks:

$$\mathbf{M} = Sigmoid(g_\phi(\mathcal{I}(\hat{\mathbf{F}}))) \in \mathbb{R}^{N \times H_l \times W_l}, \tag{5}$$

where $\mathcal{I}$ denotes bilinear interpolation, $g_\phi$ is a $1 \times 1$ convolutional block, and $Sigmoid$ denotes the sigmoid activation function. Then, the generated masks are applied to the low-level feature maps to highlight the key regions, forming the final refined fine-grained representation,

$$\hat{\mathbf{f}}_{\text{low}} = Concat(Pool(\mathbf{F}_{\text{low}}, \mathbf{M})) \in \mathbb{R}^{N \cdot C_l}, \tag{6}$$

where $Pool$ denotes masked average pooling under each of the $N$ masks, and $Concat$ represents the concatenation of resulting vectors.

**Incremental Meta-Category Optimization.** In particular, during incremental sessions, it is essential to utilize newly available data to progressively refine the model's understanding of the meta-environment (cf. **Insight 1**). At the same time, it is important to avoid significant parameter shifts—especially in scenarios with limited training data (cf. **Insight 2**).

To this end, based on Equation 4, the fine-grained feature learning process can be continuously optimized by updating the meta-category vector $\mathbf{m}$ using a limited number of training samples. During the incremental session $t'$, we compute the relation coefficients between the sample semantic representations $\mathbf{f}_{\text{high}}^i$ and the existing meta-category vector $\mathbf{m}$,

$$a_i = \mathbf{m}^\top \mathbf{f}_{\text{high}}^i / \sqrt{C_h} \in \mathbb{R}, i = 1, ..., m_t, \tag{7}$$

The final attention weights corresponding to the meta-category vector are obtained via a softmax function,

$$\hat{a}_i = \frac{\exp(a_i)}{\sum_{j=1}^{m_t} \exp(a_j)}. \tag{8}$$

Then, the meta-category vector can absorb semantic information from all training samples based on the attention weights and update itself accordingly,

$$\mathbf{m} \leftarrow (1 - \alpha)\mathbf{m} + \alpha \sum_{i=1}^{m_t} (\hat{a}_i \mathbf{f}_{\text{high}}^i) \in \mathbb{R}^{C_h}, \alpha = |\mathcal{C}^{t'}|/\sum_{t=0}^{t'} |\mathcal{C}^t|. \tag{9}$$

where $\alpha$ is a class proportion-based weight. This attention-based aggregation over new training samples further constrains the update magnitude of the meta-category vector, thereby preventing a substantial increase in the KL divergence.

### 3.3.2 Regularizing: Incremental Sub-categories Guided Feature Space Transformation

Considering the hypothesis space complexity of deep neural networks, the two complexity terms (Equations 2 and 3) can be further regularized before the final classification step (cf. **Insight 3**). On the one hand, with proper guidance, feature space transformation enables the simplification of fine-grained classification boundaries, making them easier to model with simpler classifier parameters. On the other hand, consistency regularization between fine-grained and semantic representations provides directional guidance for the transformation, encouraging a reduction in mutual information between input and representation. This helps eliminate redundant information and constrains the hypothesis space to focus on task-relevant variations.

**Sub-categories Guided Feature Space Transformation.** To retain complementary information from both the original and transformed spaces, the two type representations $\hat{\mathbf{f}}_{\text{low}}$ and $\mathbf{f}_{\text{high}}$ obtained in Section 3.3.1 are first individually mapped into a new feature space, and then fused with their original counterparts,

$$\tilde{\mathbf{f}}_{\text{low}} = (\hat{\mathbf{f}}_{\text{low}} + h_\phi(\hat{\mathbf{f}}_{\text{low}}))/2, \quad \tilde{\mathbf{f}}_{\text{high}} = (\mathbf{f}_{\text{high}} + h_\phi(\mathbf{f}_{\text{high}}))/2, \tag{10}$$

where $h_\phi$ denotes a fully connected block. During the base training phase, the overall loss function guided by sub-categories for feature extraction and feature space transformation is defined as,

$$\mathcal{L} = (1 - \beta)\mathcal{L}_{\text{high}} + \beta\mathcal{L}_{\text{low}} + \gamma\mathcal{L}_{\text{mutual}}, \tag{11}$$

where $\beta$ and $\gamma$ are hyperparameters that control the weights of each term, and $\mathcal{L}_{\text{mutual}}$ denotes the consistency regularization term. Both $\mathcal{L}_{\text{high}}$ and $\mathcal{L}_{\text{low}}$ are cross-entropy losses,

$$\mathcal{L}_* = \frac{1}{m_0} \sum_{i=1}^{m_0} -\log \Phi(\tau \mathcal{S}_{i,1:|\mathcal{C}^{(0)}|}^*), * \in \{\text{high}, \text{low}\}, \tag{12}$$

where $\mathcal{S}_{i,j}^*$ denotes cosine similarity between $\tilde{\mathbf{f}}_*^i$ and classifier $\mathbf{w}_*^j$, $\tau$ is the temperature parameter, and $\Phi$ is defined as the softmax operation,

$$\Phi(\tau \mathcal{S}_{i,1:|\mathcal{C}^{(0)}|}^*) = \frac{\exp\left(\tau \mathcal{S}_{i,y_i}^*\right)}{\sum_{j=1}^{|\mathcal{C}^{(0)}|} \exp\left(\tau \mathcal{S}_{i,j}^*\right)}. \tag{13}$$

Table 1: Performance on four fine-grained benchmark datasets. Due to space limitations, only the accuracy of the final session ($\text{Acc}_T$) and average accuracy across all sessions ($\text{Acc}_{Avg}$) are presented.

| Method | CUB200 | | Stanford Dogs | | Stanford Cars | | FGVCAircraft | |
|---|---|---|---|---|---|---|---|---|
| | $\text{Acc}_T$ | $\text{Acc}_{Avg}$ | $\text{Acc}_T$ | $\text{Acc}_{Avg}$ | $\text{Acc}_T$ | $\text{Acc}_{Avg}$ | $\text{Acc}_T$ | $\text{Acc}_{Avg}$ |
| CEC [32] | 34.50 | 43.71 | 32.65 | 40.56 | 37.09 | 48.24 | 29.57 | 36.94 |
| C-FSCIL [7] | 13.86 | 17.62 | 16.68 | 22.78 | 18.90 | 27.76 | 28.28 | 38.60 |
| FACT [37] | 32.71 | 42.50 | 33.26 | 42.96 | 43.98 | 56.17 | 31.60 | 40.46 |
| TEEN [26] | 30.08 | 39.33 | 39.14 | 47.86 | 45.32 | 57.37 | 30.98 | 43.42 |
| CLOSER [17] | 31.74 | 43.95 | 34.37 | 44.69 | 34.49 | 53.78 | 36.56 | 48.08 |
| PFR [18] | 33.99 | 43.70 | 36.01 | 44.25 | 44.56 | 58.12 | 34.42 | 43.05 |
| ADBS [11] | 27.85 | 38.71 | 29.15 | 41.54 | 29.79 | 46.50 | 35.42 | 45.23 |
| **Ours** | **36.00** | **45.44** | **44.06** | **52.38** | **57.58** | **68.43** | **40.72** | **54.17** |

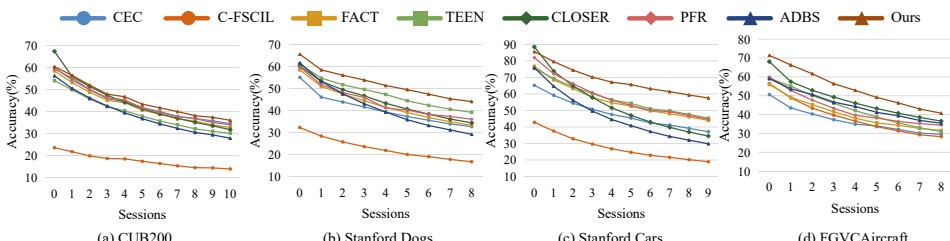

Figure 3: Classification accuracy for each session on four fine-grained benchmark datasets.

**Incremental Sub-category Relationship Optimization.** Meanwhile, $\mathcal{L}_{\text{mutual}}$ is used to continuously constrain the sub-category relationship distributions between the fine-grained and semantic representations,

$$\mathcal{L}_{\text{mutual}} = \frac{1}{m_0} \sum_{i=1}^{m_0} \left[ D_{\text{KL}} \left( \Phi(\tau \mathcal{S}_{i,1:|\mathcal{C}^{(0)}|}^{\text{low}}) \parallel \Phi(\tau \mathcal{S}_{i,1:|\mathcal{C}^{(0)}|}^{\text{high}}) \right) \right.$$
$$\left. + D_{\text{KL}} \left( \Phi(\tau \mathcal{S}_{i,1:|\mathcal{C}^{(0)}|}^{\text{high}}) \parallel \Phi(\tau \mathcal{S}_{i,1:|\mathcal{C}^{(0)}|}^{\text{low}}) \right) \right] \tag{14}$$

where $D_{\text{KL}}(p \parallel q) = \sum_j p_j \log p_j / q_j$. After the base training phase, as in most FSCIL methods, the classifier is replaced with prototype-based classifiers and can be extended during the incremental session to accommodate novel classes, $\mathbf{w}_*^c = \frac{1}{Num_c} \sum_{y_i=c} \tilde{\mathbf{f}}_*^i$, where $Num_c$ denotes the number of training samples for class $c$. Therefore, during the incremental training phase, the mutual learning regularization term $\mathcal{L}_{\text{mutual}}$ can be computed on the new training data to maintain and refine the novel sub-category relationship distributions, enabling ongoing interaction and refinement between the two types of sub-category prototype representations. This regularization not only constrains the feature space used for classification but also encourages mutual complementarity between the two types of representations, thereby implicitly integrating fine-grained knowledge into the semantic representations.

Thus, the classification inference in session $t'$ can be performed using only $\tilde{\mathbf{f}}_{\text{high}}$,

$$y_i^\star = \underset{c \in \cup_{t=0}^{t'} \mathcal{C}^t}{\arg\max} \mathcal{S}_{i,1:\sum_{t=0}^{t'} |\mathcal{C}^{(t)}|}^{\text{high}}. \tag{15}$$

## 4 Experiments

### 4.1 Experimental Setup

**Datasets.** Following the benchmark setting in [18], we evaluate our method on four fine-grained datasets: CUB200 [25], Stanford Dogs [8], Stanford Cars [30], and FGVCAircraft [15]. The data split details are provided in Appendix A.3.

Table 2: Ablation studies of our proposed method on the Stanford Dogs dataset.

| Model | Infer | Accuracy in each session (%) | | | | | | | | |
|---|---|---|---|---|---|---|---|---|---|---|
| | | 0 | 1 | 2 | 3 | 4 | 5 | 6 | 7 | 8 |
| $M_0$ | $\mathcal{S}^{\text{high}}$ | 62.78 | 55.10 | 51.55 | 48.68 | 45.25 | 42.15 | 40.12 | 38.23 | 36.68 |
| $M_1$ | $\mathcal{S}^{\text{high}}\mathcal{S}^{\text{low}}$ | 65.58 | 57.32 | 53.82 | 51.08 | 48.07 | 45.93 | 43.45 | 41.45 | 39.82 |
| $M_2$ | $\mathcal{S}^{\text{high}}\mathcal{S}^{\text{low}}$ | **66.51** | 58.37 | 54.70 | 52.05 | 49.08 | 46.79 | 44.44 | 42.28 | 40.63 |
| $M_3$ | $\mathcal{S}^{\text{high}}\mathcal{S}^{\text{low}}$ | **66.51** | 58.40 | 55.12 | 52.59 | 49.73 | 47.70 | 45.37 | 43.22 | 41.50 |
| $M_4$ | $\mathcal{S}^{\text{high}}\mathcal{S}^{\text{low}}$ | 66.16 | 58.27 | 54.63 | 52.06 | 49.67 | 47.40 | 45.05 | 43.28 | 41.60 |
| $M_5$ | $\mathcal{S}^{\text{high}}\mathcal{S}^{\text{low}}$ | 65.78 | **58.44** | **56.03** | 53.48 | 50.87 | 48.95 | 46.97 | 44.68 | 43.36 |
| $M_5$ | $\mathcal{S}^{\text{low}}$ | 65.03 | 57.68 | 55.40 | 52.72 | 49.97 | 48.03 | 45.98 | 43.67 | 42.30 |
| $M_5$ | $\mathcal{S}^{\text{high}}$ | 65.57 | **58.44** | **56.03** | **53.84** | **51.34** | **49.45** | **47.42** | **45.29** | **44.06** |

**Experimental details.** Following [18], we employ ResNet-12 as the backbone. In our experiments, $\mathbf{F}_{\text{high}}$ is the output of the final layer of the backbone, while $\mathbf{F}_{\text{low}}$ is the output of the penultimate layer. Following previous works [18], the temperature hyperparameter $\tau$ is set to 16. $N$ is set as an integer 4 to ensure diversity in fine-grained features and kept small for efficiency. $\beta$ and $\gamma$ are set to 0.5 to balance the optimization strength among different components. For comparative methods whose results are not reported in [18], we reproduce their performance under the same experimental settings using the publicly available source code. Please see Appendix A.3 for details.

## 4.2 Comparison Results

In this section, we compare our method with recent FSCIL methods and FG-FSCIL method in Table 1 and Figure 3. It can be observed that our method achieves the best overall performance. Specifically, our method achieves a 1.21% higher average accuracy ($\text{Acc}_{Avg}$) than CLOSER on the CUB200 dataset, and notably outperforms it by 4.26% in the final session ($\text{Acc}_T$). This suggests that our method, by progressively evolving fine-grained feature extraction while constraining hypothesis space complexity, effectively improves performance on incremental classes with limited samples, often leading to a more pronounced gain in the final session. Although ADBS considers model adaptation during the incremental sessions, it lacks mechanisms for inter-class synergy modeling and hypothesis space complexity regularization, resulting in limited performance. Detailed experimental results can be found in Appendix A.4.

## 4.3 Ablation Study

To demonstrate the effectiveness of each component in our method, we conduct ablation experiments in Table 2. $\mathcal{S}^{\text{high}}$ denotes using only high-level features during inference, as described in Equation 15, while $\mathcal{S}^{\text{low}}$ denotes using only low-level features. $\mathcal{S}^{\text{high}}\mathcal{S}^{\text{low}}$ represents using the sum of their similarity scores. $M_0$ represents the baseline.

(1) Firstly, $M_1$ applies fine-grained feature extraction without the meta-category vector (i.e., under Equations 5-6, $\hat{\mathbf{F}}$ is directly set to $\mathbf{F}_{\text{high}}$), together with the basic cross-entropy loss (Equation 12), yields an initial performance improvement. (2) $M_2$ builds on $M_1$ by introducing the meta-category vector (Equation 4), (3) and $M_3$ further performs incremental optimization of this meta-category vector (Equations 7-9), leading to additional performance gains. Further analysis in Section 4.4.1 demonstrates the effectiveness of our optimization strategy. The improved fine-grained feature extraction benefits all sub-categories within the meta-category environment, yielding relatively consistent performance improvements across sessions. (4) $M_4$ adds feature space transformation (Equation 10) to $M_3$ does not lead to significant performance gains. (5) However, $M_5$, which further includes sub-category relationship regularization (Equation 14), shows clear improvement. This indicates that regularizing the alignment between the two relationship distributions enables the feature space transformation process to eliminate redundancy while preserving classification-relevant information, effectively reducing hypothesis space complexity and, consequently, the two complexity terms. This reduction is especially beneficial for classes with fewer training samples, leading to more

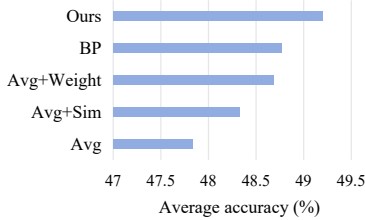

Figure 4: Analysis of meta-category optimization strategies.

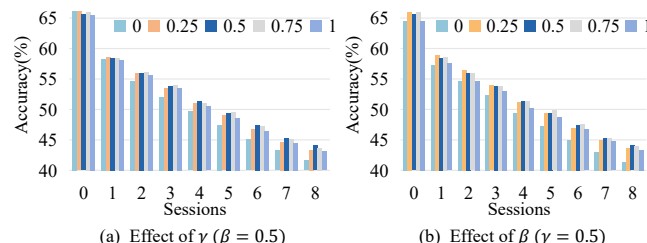

(a) Effect of $\gamma$ ($\beta = 0.5$)

(b) Effect of $\beta$ ($\gamma = 0.5$)

Figure 5: The influence of hyperparameters $\gamma$ and $\beta$.

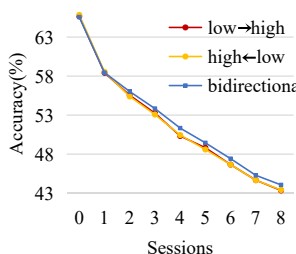

Figure 6: Analysis of $\mathcal{L}_{\text{mutual}}$.

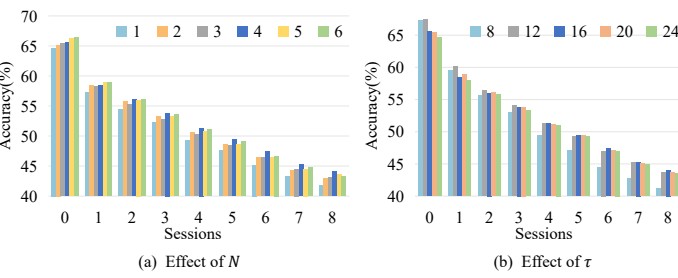

(a) Effect of $N$

(b) Effect of $\tau$

Figure 7: The influence of hyperparameters $N$ and $\tau$.

noticeable performance improvements in the later incremental sessions. (6) Besides, regularization allows inference using only high-level features ($\mathcal{S}^{\text{high}}$) to achieve performance comparable to or better than using both ($\mathcal{S}^{\text{high}}\mathcal{S}^{\text{low}}$). Further analysis in Section 4.4.2 demonstrates the impact of fine-grained feature extraction and regularization on class representations.

## 4.4 Further Analysis

### 4.4.1 Different Meta-Category Optimization Strategies

To demonstrate the effectiveness of our meta-category optimization strategy (Equations 7-9 in Section 3.3.1), we compare the average accuracy across incremental sessions on the Stanford Dogs dataset under different optimization schemes, as shown in Figure 4.

It can be observed that directly updating the meta-category vector $\mathbf{m}$ by averaging it with the feature vectors $\mathbf{f}_{\text{high}}^{i}$ of all training samples (**Avg**) leads to poor performance. Introducing cosine similarity as a soft weight between $\mathbf{m}$ and the mean of $\mathbf{f}_{\text{high}}^{i}$ (**Avg+Sim**) slightly limits the update magnitude but still performs poorly. This suggests that even with few parameters, optimization under limited samples can easily cause posterior drift — a key reason why existing FG-FSCIL methods do not incorporate new class knowledge into the feature extractor. Replacing cosine similarity with a class proportion-based weight $\alpha$ (**Avg+Weight**) helps reduce the update magnitude in later sessions, leading to moderate improvements. Our proposed method (**Ours**, Equation 9) first computes a cosine-weighted sum of all $\mathbf{f}_{\text{high}}^{i}$ to mitigate the influence of outliers, then applies $\alpha$ to further constrain the impact of new classes—yielding the best performance. While backpropagation-based optimization (**BP**) also perform well, they require careful tuning of learning rate and training epochs. In contrast, our strategy achieves effective updates with no additional hyperparameters.

### 4.4.2 Effect of Terms $\mathcal{L}_{\text{mutual}}$ and $\mathcal{L}_{\text{low}}$

To analyze the effects of the regularization term $\mathcal{L}_{\text{mutual}}$ and the fine-grained feature extraction term $\mathcal{L}_{\text{low}}$ (Section 3.3.2), we evaluate the influence of their corresponding hyperparameters $\gamma$ and $\beta$ (Equation 11) on the Stanford Dogs dataset, as shown in Figure 5.

Figure 5(a) shows that in session 0, the value of $\gamma$ (i.e., whether the regularization term is applied) has little effect on performance. This suggests that for base classes with ample training samples (large $m_0$), high-level semantic features already generalize well, and regularization has limited impact on the complexity term (Equation 2). However, as subsequent sessions introduce more new

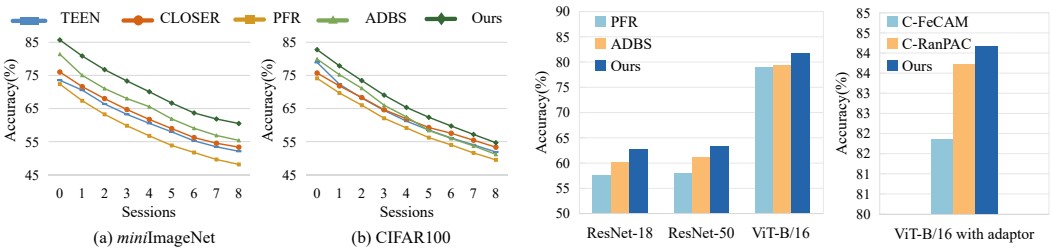

Figure 8: Classification accuracy on conventional FSCIL benchmarks.

Figure 9: Accuracy after the last session on CUB200 with different backbones.

classes with limited training samples, the advantage of regularization gradually emerges, consistently outperforming the no-regularization setting across different values. Figure 5(b) shows that applying fine-grained feature extraction ($\beta > 0$) consistently improves performance across all sessions. This is particularly evident in later sessions with more incremental classes, where limited training samples lead to less reliable semantic representations, making fine-grained features more critical for recognition—i.e., a larger $\beta$ is needed. Optimal performance is generally achieved when $\gamma$ and $\beta$ are set around 0.5. Furthermore, we analyze the difference between using bidirectional KL divergence (i.e., $\mathcal{L}_{\text{mutual}}$) and unidirectional KL divergence (i.e., only high $\rightarrow$ low or only low $\rightarrow$ high). As shown in Figure 6, bidirectional KL divergence leads to better performance. Our approach adopts bidirectional KL divergence to enable mutual regularization and refinement between the two branches, thereby achieving better alignment of their sub-category relationship distributions.

### 4.4.3 Effect of Hyperparameters $N$ and $\tau$

In this section, we analyze the effect of other hyperparameters, the number of attention masks $N$ and the temperature $\tau$. The experimental results are shown in Figure 7.

It can be observed that MEL's performance remains relatively stable across a range of values for these hyperparameters. When $N \geq 4$, increasing $N$ yields no additional performance gain, with $N = 4$ achieving the best overall performance, particularly in the later sessions. Additionally, consistent with other FSCIL methods, our MEL demonstrates sustained performance advantages when $\tau = 16$. These results are consistent with the principles behind our chosen hyperparameter settings.

### 4.4.4 Performance on Other FSCIL Benchmarks and Backbones

Although the analysis primarily focuses on fine-grained datasets and the proposed method is tailored for FG-FSCIL, it still outperforms recent methods on conventional FSCIL benchmarks [22, 9], as shown in Figure 8. This is because the generalization error bound analysis for FSCIL remains valid. While fine-grained feature extraction is not crucial for conventional datasets, constraining hypothesis space complexity during empirical error minimization is still essential. Moreover, our method still achieves better performance than other approaches (e.g., PFR [18], ADBS [11], and C-FeCAM/C-RanPAC [6]) across different backbones, as shown in Figure 9.

## 5 Conclusion

In conclusion, we revisit the Fine-Grained Few-Shot Class-Incremental Learning (FG-FSCIL) problem from a theoretical perspective and formulate it within a shared meta-environment. By deriving a generalization error bound, we demonstrate the importance of further minimizing the complexity of both per-session sub-categories and the overall meta-environment. Building upon these insights, we propose the Meta-Environment Learner (MEL), which progressively evolves fine-grained feature representations while regularizing hypothesis space complexity. Extensive experiments on benchmark datasets validate the effectiveness of our approach, showing consistent and significant improvements over state-of-the-art methods.

## Acknowledgments and Disclosure of Funding

This work was supported in part by the National Natural Science Foundation of China under Grant 62202272, 62172256, 62202278, 62302276, in part by the Natural Science Foundation of Shandong Province under Grant ZR2024LZH002 and Taishan Scholar Project of Shandong Province under Grant tstp20250704.

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

# A  Technical Appendices and Supplementary Material

## A.1  Proof of the Generalization Bound for FSCIL

In this section, we prove Theorem 3.1 utilizing classical PAC-Bayes bound.

**Theorem A.1** (PAC-Bayes Bound [16]). *Let $\mathcal{X}$ be a sample space and $\mathbb{X}$ a distribution over $\mathcal{X}$, and let $\mathcal{F}$ be a hypothesis class of functions on $\mathcal{X}$. Let $g : \mathcal{F} \times \mathcal{X} \to [0,1]$ be a loss function. Suppose $X_1, \ldots, X_K$ are i.i.d. samples from $\mathbb{X}$ and let $\pi$ be a prior distribution over $\mathcal{F}$ (independent of the samples). Then for any $\delta \in (0,1]$, the following holds uniformly for all posterior distributions $\rho$ over $\mathcal{F}$ (even sample dependent) with probability at least $1 - \delta$:*

$$\mathbb{E}_{f \sim \rho}\mathbb{E}_{X \sim \mathbb{X}}[g(f, X)] \leq \frac{1}{K}\sum_{k=1}^{K}\mathbb{E}_{f \sim \rho}[g(f, X_k)] + \sqrt{\frac{1}{2(K-1)}\left(D_{\mathrm{KL}}(\rho\|\pi) + \log\frac{K}{\delta}\right)}. \quad (16)$$

*where $D_{\mathrm{KL}}(\rho\|\pi)$ denotes the Kullback–Leibler (KL) divergence between the posterior distribution $\rho$ and the prior distribution $\pi$, $D_{\mathrm{KL}}(Q\|P) \triangleq \mathbb{E}_{h \sim Q} \log \frac{Q(h)}{P(h)}$.*

*Proof.* According to Theorem A.1, we analyze the generalization error for the classes introduced in each session. The samples are $X_K \triangleq z_{t,i}$, $K \triangleq m_t$, and their distribution is $\mathbb{X} \triangleq \mathcal{D}_t$. Formally, we set $\rho = Q_{t'}$ and $\pi = P_t$. We get that for any $\delta_t > 0$, with probability at least $1 - \delta_t$ over $S_{train}^t \sim \mathcal{D}_t^{m_t}$,

$$er(Q_{t'}, \mathcal{D}_t) \leq \widehat{er}(Q_{t'}, S_{train}^t) + \sqrt{\frac{1}{2(m_t - 1)}\left(D_{\mathrm{KL}}(Q_{t'}\|P_t) + \log\frac{m_t}{\delta_t}\right)}, \quad (17)$$

where $P_t \triangleq Q_{t-1}$ and $Q_{t'}$ represents the posterior distribution in session $t'$, i.e., $Q_{t'} \triangleq Q(S_{t'}, P_{t'})$.

Then, we consider the generalization across $\mathcal{D}_t^{m_t}$ from different sessions, thereby providing a bound on the environment-level generalization. Let $(\mathcal{D}_t, m_t, S_{train}^t)$ for $t = 0, 1, \ldots, t'$ be drawn i.i.d. from the environment $\mathcal{T}$ and $S_{train}^t \sim \mathcal{D}_t^{m_t}$. Using Theorem A.1 again, let the prior be $\pi = P_0$ and posterior $\rho = Q_{t'}$. Then, for any $\delta_0 > 0$, with probability at least $1 - \delta_0$,

$$er(Q_{t'}, \mathcal{T}) \leq \sum_{t=0}^{t'}\alpha_t er(Q_{t'}, \mathcal{D}_t) + \sqrt{\frac{1}{2t'}\left(D_{\mathrm{KL}}(Q_{t'}\|P_0) + \log\frac{t'+1}{\delta_0}\right)}, \quad (18)$$

where $\alpha_t = \frac{|\mathcal{C}^t|}{\sum_{i=0}^{t'}|\mathcal{C}^i|}$, reflecting the relationship among the class sets introduced in different sessions.

Finally, to bound the probability of the intersection of the events defined in Equations 17 and 18, we make use of the union bound. Let $\delta > 0$, set $\delta_0 \triangleq \frac{\delta}{2}$ and $\delta_t \triangleq \frac{\delta}{2(t'+1)\alpha_t}$ for $t = 0, 1, ..., t'$. By applying the union bound argument (Lemma A.2), with probability at least $1 - \delta$,

$$er(Q_{t'}, \mathcal{T}) \leq \sum_{t=0}^{t'}\alpha_t\widehat{er}(Q_{t'}, S_{train}^t) \quad (19)$$

$$+ \sum_{t=0}^{t'}\alpha_t\sqrt{\frac{1}{2(m_t - 1)}\left(D_{\mathrm{KL}}(Q_{t'}\|P_t) + \log\frac{m_t}{\delta_t}\right)} \quad (20)$$

$$+ \sqrt{\frac{1}{2t'}\left(D_{\mathrm{KL}}(Q_{t'}\|P_0) + \log\frac{t'+1}{\delta_0}\right)}, \quad (21)$$

where $D_{\mathrm{KL}}(Q_{t'}\|P_t) = \sum_{i=t}^{t'}D_{\mathrm{KL}}(Q_i\|P_i)$. The probability is taken over sampling of $(\mathcal{D}_t, m_t) \sim \mathcal{T}$ and $S_{train}^t \sim \mathcal{D}_t^{m_t}, t = 0, 1, ..., t'$. $\qquad \square$

In the following, we present the statement of Lemma A.2 along with its proof.

**Lemma A.2.** *Let $\{E_i\}_{i=1}^n$ be a set of events such that $\mathbb{P}(E_i) \geq 1 - \delta_i$, for some $\delta_i \geq 0$, $i = 1, \ldots, n$. Then,*

$$\mathbb{P}\left(\bigcap_{i=1}^n E_i\right) \geq 1 - \sum_{i=1}^n \delta_i.$$

*Proof.* We begin by observing that

$$\mathbb{P}\left(\bigcap_{i=1}^n E_i\right) = 1 - \mathbb{P}\left(\bigcup_{i=1}^n E_i^c\right), \tag{22}$$

where $E_i^c$ denotes the complement of event $E_i$.

By applying the union bound, we obtain

$$\mathbb{P}\left(\bigcup_{i=1}^n E_i^c\right) \leq \sum_{i=1}^n \mathbb{P}(E_i^c) = \sum_{i=1}^n \left(1 - \mathbb{P}(E_i)\right). \tag{23}$$

Using the assumption $\mathbb{P}(E_i) \geq 1 - \delta_i$, we have

$$\sum_{i=1}^n \left(1 - \mathbb{P}(E_i)\right) \leq \sum_{i=1}^n \delta_i. \tag{24}$$

Combining the above results gives

$$\mathbb{P}\left(\bigcap_{i=1}^n E_i\right) \geq 1 - \sum_{i=1}^n \delta_i. \tag{25}$$

$\square$

## A.2 Limitations and Future Work

Following the conventional FG-FSCIL setting, our method assumes base and novel classes are sampled from the same domain. This is a standard and practical choice in prior studies, ensuring consistency in semantic granularity. Nevertheless, real-world scenarios may involve more diverse data sources, such as photographs and sketches. Extending the framework to cross-domain FG-FSCIL is a natural and worthwhile extension, which we plan to explore in future work.

## A.3 Detailed Experimental Setups

**Datasets.** For CUB200, 100 classes are used for base training, and the remaining 100 classes are split into 10 incremental sessions. For Stanford Dogs, 80 classes are used in the base session, and the remaining 40 classes are evenly divided into 8 incremental sessions. Each incremental class contains 5 samples, forming a 5-way 5-shot FSCIL setting. For Stanford Cars, 106 classes are used for base training, and the remaining 90 classes are split across 9 sessions, following a 10-way 5-shot setting. For FGVCAircraft, 60 classes are used for base training and 40 for incremental learning. A 5-way 5-shot setting is applied, resulting in 9 sessions in total.

**Experimental details.** Our implementation is based on the code released by PFR [18] under the MIT license. Our method is conducted with PyTorch library on a single NVIDIA 3090, and SGD with momentum is used for optimization. All method-agnostic hyperparameter settings are kept consistent with those of PFR.

## A.4 Detailed Experimental Results and Analysis

Table 3: Classification accuracy on the CUB200 dataset.

| Method | Accuracy in each session (%) | | | | | | | | | | |
|--------|-------|-------|-------|-------|-------|-------|-------|-------|-------|-------|-------|
| | 0 | 1 | 2 | 3 | 4 | 5 | 6 | 7 | 8 | 9 | 10 |
| CEC | 59.94 | 54.31 | 49.88 | 45.81 | 44.44 | 41.72 | 39.95 | 37.74 | 36.79 | 35.74 | 34.50 |
| C-FSCIL | 23.53 | 21.73 | 19.83 | 18.62 | 18.45 | 17.34 | 16.36 | 15.27 | 14.48 | 14.34 | 13.86 |
| FACT | 58.45 | 53.14 | 48.77 | 45.05 | 43.97 | 40.57 | 38.77 | 36.62 | 35.60 | 33.86 | 32.71 |
| TEEN | 53.98 | 49.86 | 45.69 | 42.19 | 40.32 | 37.84 | 35.65 | 33.92 | 32.10 | 31.02 | 30.08 |
| CLOSER | **67.35** | 55.90 | 51.37 | 47.37 | 44.22 | 41.22 | 38.86 | 36.94 | 35.04 | 33.46 | 31.74 |
| PFR | 59.32 | 55.19 | 50.03 | 46.26 | 45.24 | 41.59 | 39.64 | 37.72 | 36.75 | 35.01 | 33.99 |
| ADBS | 56.21 | 50.47 | 46.14 | 42.50 | 39.39 | 36.71 | 34.36 | 32.30 | 30.47 | 29.36 | 27.85 |
| Ours | 60.40 | **56.43** | **52.00** | **47.88** | **46.69** | **43.34** | **41.65** | **39.96** | **38.07** | **37.37** | **36.00** |

Table 4: Classification accuracy on the Stanford Dog dataset.

| Method | Accuracy in each session (%) | | | | | | | | |
|--------|-------|-------|-------|-------|-------|-------|-------|-------|-------|
| | 0 | 1 | 2 | 3 | 4 | 5 | 6 | 7 | 8 |
| CEC | 55.01 | 46.05 | 43.83 | 41.57 | 39.28 | 37.26 | 35.53 | 33.87 | 32.65 |
| C-FSCIL | 32.22 | 28.34 | 25.68 | 23.57 | 21.81 | 20.00 | 19.00 | 17.74 | 16.68 |
| FACT | 58.43 | 50.92 | 47.48 | 44.39 | 41.48 | 39.04 | 36.78 | 34.87 | 33.26 |
| TEEN | 61.09 | 54.70 | 51.71 | 49.56 | 47.28 | 44.41 | 42.30 | 40.56 | 39.14 |
| CLOSER | 60.22 | 53.22 | 49.40 | 46.64 | 43.24 | 40.65 | 38.33 | 36.10 | 34.37 |
| PFR | 59.78 | 51.65 | 48.14 | 45.91 | 41.13 | 40.13 | 38.20 | 37.28 | 36.01 |
| ADBS | 61.63 | 53.15 | 47.55 | 43.02 | 39.28 | 35.78 | 33.16 | 31.16 | 29.15 |
| Ours | **65.57** | **58.44** | **56.03** | **53.84** | **51.34** | **49.45** | **47.42** | **45.29** | **44.06** |

Table 5: Classification accuracy on the Stanford Cars dataset.

| Method | Accuracy in each session (%) | | | | | | | | | |
|--------|-------|-------|-------|-------|-------|-------|-------|-------|-------|-------|
| | 0 | 1 | 2 | 3 | 4 | 5 | 6 | 7 | 8 | 9 |
| CEC | 65.23 | 59.22 | 54.43 | 50.75 | 47.57 | 45.38 | 42.41 | 41.16 | 39.17 | 37.09 |
| C-FSCIL | 42.80 | 37.50 | 32.83 | 29.63 | 26.76 | 24.64 | 22.82 | 21.57 | 20.18 | 18.90 |
| FACT | 76.76 | 68.57 | 63.29 | 57.93 | 54.59 | 52.69 | 49.77 | 48.02 | 46.12 | 43.98 |
| TEEN | 75.42 | 69.39 | 64.30 | 60.24 | 56.44 | 54.32 | 50.96 | 49.78 | 47.56 | 45.32 |
| CLOSER | **88.56** | 73.75 | 64.85 | 57.85 | 51.58 | 47.01 | 42.98 | 39.78 | 36.92 | 34.49 |
| PFR | 82.02 | 72.22 | 66.19 | 60.73 | 56.19 | 53.04 | 50.00 | 49.22 | 47.02 | 44.56 |
| ADBS | 75.91 | 64.64 | 56.16 | 49.65 | 44.49 | 40.79 | 37.24 | 34.26 | 32.02 | 29.79 |
| Ours | 85.63 | **79.62** | **74.46** | **70.24** | **67.17** | **65.65** | **63.15** | **61.41** | **59.36** | **57.58** |

Table 6: Classification accuracy on the FGVCAircraft dataset.

| Method | Accuracy in each session (%) | | | | | | | | |
|--------|-------|-------|-------|-------|-------|-------|-------|-------|-------|
| | 0 | 1 | 2 | 3 | 4 | 5 | 6 | 7 | 8 |
| CEC | 50.61 | 43.50 | 40.32 | 37.26 | 34.93 | 33.95 | 32.15 | 30.17 | 29.57 |
| C-FSCIL | 56.45 | 48.84 | 43.40 | 39.71 | 36.52 | 33.62 | 31.33 | 29.28 | 28.28 |
| FACT | 55.95 | 49.19 | 45.25 | 41.53 | 38.06 | 35.58 | 34.31 | 32.64 | 31.60 |
| TEEN | 59.17 | 55.08 | 50.06 | 46.05 | 42.10 | 38.73 | 35.60 | 33.01 | 30.98 |
| CLOSER | 68.07 | 57.45 | 52.86 | 49.28 | 46.05 | 43.20 | 40.73 | 38.51 | 36.56 |
| PFR | 59.94 | 52.80 | 47.66 | 43.32 | 39.61 | 38.16 | 36.36 | 35.22 | 34.42 |
| ADBS | 59.03 | 53.67 | 50.60 | 46.61 | 44.28 | 41.19 | 39.36 | 36.90 | 35.42 |
| Ours | **71.43** | **66.30** | **61.66** | **56.34** | **52.85** | **49.09** | **46.18** | **42.96** | **40.72** |

