# OpenReview forum: "Evolving and Regularizing Meta-Environment Learner for Fine-Grained Few-Shot Class-Incremental Learning"
_NeurIPS.cc/2025/Conference — NeurIPS 2025 poster_

### Official Review · Reviewer_zZGu · 2025-07-01

**Clarity:** 3
**Significance:** 3
**Originality:** 3
**Rating:** 5
**Confidence:** 5

**Summary:**

The paper proposes a novel Meta-Environment Learner (MEL) for Fine-Grained Few-Shot Class-Incremental Learning (FG-FSCIL). Key contributions include:
1.	Theoretical formulation of FSCIL problem with derived generalization error bounds
2.	MEL framework with evolving feature extraction and hypothesis space regularization
3.	Comprehensive experiments showing state-of-the-art performance on fine-grained benchmarks

**Questions:**

1.	Why not use a dual branch approach for inference when performing dual branch calculations on both high-level and low-level features during training?
2.	Is the fully linked module used for high-level and low-level features in formula (10) shared? Why can we do this?
3.	Why does Eq (10) input features into a fully linked module, and what space do you expect the features to be transformed into?
4.	Many previous FSCIL works have also been experimented on fine-grained data (CUB200), although you used different backbones (resnet12 vs resnet18), I suggest you also use the same settings for comparison, which will greatly enhance the persuasiveness of this submission.
5.	In the experimental section, it is recommended to adjust the initial accuracy to the same level for fair comparison.

**Ethical Concerns:**

["NO or VERY MINOR ethics concerns only"]

**Limitations:**

Yes

**Quality:**

3

**Strengths And Weaknesses:**

Strengths
•	The generation bound (Theorem 3.1) provides solid theoretical foundation.
•	The thinking on high and low features has important insights and potential impact on promoting progress in the FSCIL field
•	Extensive Validation: Experiments on 4 fine-grained datasets demonstrate consistent improvements.
Weaknesses
•	Computational Cost: Additional overhead from meta-category optimization not quantified
•	Error Analysis: Lacks case studies on failure modes or challenging categories

---

> ### Author Rebuttal · Authors · 2025-07-30
>
> We sincerely appreciate the constructive feedback. Detailed responses to the weaknesses and questions are as follows.
>
> **Response to Weaknesses**
>
> **1.** Compared to other training steps involving backpropagation, the computational cost of meta-category optimization is negligible, as it involves only forward-pass arithmetic operations and does not require any backpropagation or gradient updates. We manually estimate the total cost to be approximately 0.00013G. This computational cost is too small and is not reflected in empirical timing results.
>
> **2.** Due to rebuttal format restrictions, we are unable to include failure case pictures here, but we will add failure cases and corresponding error analysis in the supplementary material.
>
> **Response to Questions**
>
> **1.** During training, we introduce a regularization term (Eq. (14)) that encourages interaction and mutual refinement between the learned representations of the two branches to achieve alignment (L230-233). Thus, a single branch, particularly the high-level branch with its semantic focus, tends to yield comparable or better performance than dual-branch inference (Table 2, last three rows), while improving inference efficiency.
>
> **2.** They are not shared. We will revise Eq. (10) to clearly distinguish them. Since the input features differ between the two branches, separate fully linked modules are required.
>
> **3.** The fully connected module is used to individually transform high-level and low-level features guided by the loss function.
>
> Classification loss term (Eq. (12)) separately guides two types of features to be transformed into class-discriminative feature spaces. Meanwhile, mutual learning regularization term (Eq. (14)) regularizes two types of features to align in feature spaces with consistent sub-category similarity relationships. Thus, during the transformation process, Eq. (12) preserves classification-relevant information, while Eq. (14) eliminates redundant information irrelevant to modeling category similarity. This simplifies the classification and further reduces the generalization bound by lowering hypothesis space complexity (L203-210). Experimental results also show that transformation with proper guidance leads to performance improvements (Table 2).
>
> **4.** We follow the existing FG-FSCIL setting by using ResNet12 as the backbone. As suggested, we also provide comparison results using ResNet18 on CUB200, as shown below. Our method still outperforms recent methods under this setting. Thanks for the suggestion. We will include these results in the paper to enhance its persuasiveness.
>
> |       |   PFR   |  ADBS  |  Ours  |
> |-------|:-------:|:------:|:------:|
> | $\text{Acc}_T$  |  57.54  | 60.14  | 62.84  |
> | $\text{Acc}_{Avg}$|  65.64  | 67.75  | 69.20  |
>
> **5.** Different methods introduce distinct modules during the base session, which inevitably leads to varying initial accuracies. This is a common phenomenon in FSCIL research. Given that fair comparison is important, we have kept all method-agnostic settings as consistent as possible and achieved initial accuracy close to that of some existing methods (e.g., PFR on CUB200). Moreover, our experiments include comparison methods with both higher and lower initial accuracies than ours. In all these cases, our method consistently performs better in subsequent sessions (Figure 3), which ensures a fair comparison and demonstrates its effectiveness in the incremental learning setting.

---

> > ### Comment · Reviewer_zZGu · 2025-08-06
> > **Comment by reviewers**
> >
> > Thanks for the response and the rebuttal, which addresses my concerns.

---

### Official Review · Reviewer_VXZc · 2025-07-02

**Clarity:** 3
**Significance:** 3
**Originality:** 3
**Rating:** 4
**Confidence:** 2

**Summary:**

This paper addresses the Fine-Grained Few-Shot Class-Incremental Learning (FG-FSCIL) problem by proposing a novel Meta-Environment Learner (MEL). The authors provide a theoretical formulation of FSCIL and derive generalization error bounds within a shared fine-grained meta-category environment. Based on these theoretical insights, MEL is designed to evolve fine-grained feature extraction capabilities while regularizing hypothesis space complexity. The method introduces an incrementally optimized meta-category vector and sub-category relationship regularization to achieve better performance on fine-grained incremental learning tasks.

**Questions:**

(1) How does the method perform when the base and incremental classes come from different domains or have varying granularity levels? (2) Could you provide more intuition about why the attention-based aggregation in Eq. 9 is superior to other optimization strategies? The experimental comparison in Figure 4 is helpful but lacks theoretical justification. (3) The paper claims that fine-grained categories naturally foster "latent synergy" - could you provide empirical evidence or visualization to support this claim? (4) In Equation 14, why use bidirectional KL divergence instead of unidirectional? What's the empirical difference?

**Ethical Concerns:**

["NO or VERY MINOR ethics concerns only"]

**Final Justification:**

I have read the feedback. I will keep the score.

**Limitations:**

Yes

**Paper Formatting Concerns:**

No.

**Quality:**

3

**Strengths And Weaknesses:**

Strengths: (1) The paper provides a rigorous theoretical analysis of the FSCIL problem, deriving generalization bounds that offer meaningful insights into the learning process. The connection between theory and method design is well-established. (2) The concept of leveraging the meta-environment and inter-class synergy in fine-grained data is innovative and well-motivated. This addresses a genuine limitation in existing FG-FSCIL methods. (3) The experimental evaluation is thorough, covering four fine-grained datasets with consistent improvements over state-of-the-art methods. The ablation studies effectively demonstrate the contribution of each component.

Weaknesses: (1) While the theoretical motivation is strong, the actual implementation relies heavily on standard techniques (attention mechanisms, KL divergence regularization). The meta-category vector optimization, though effective, is relatively straightforward. (2) The theoretical section jumps quickly into complex bounds without sufficient intuitive explanation. Some key concepts like "meta-environment" could be introduced more gradually.

---

> ### Author Rebuttal · Authors · 2025-07-30
>
> We sincerely appreciate the constructive feedback. Detailed responses to the weaknesses and questions are as follows.
>
> **Response to Weaknesses**
>
> **1.** The primary goal of this paper is to uncover and explore the significance of fine-grained inter-class synergy in FSCIL from the perspective of meta-environment, supported by both theoretical analysis and empirical validation. Using standard techniques and straightforward implementations helps focus on the theoretical insights and motivation itself, and strengthens the connection between theory and method design. Experimental results demonstrate that, guided by our theoretical motivation, the thoughtful design based on these techniques has achieved strong performance. This underscores the significance of our theoretical insights and highlights their potential for further research.
>
>
> **2.** We define the concept of meta-environment in Definition 3.2 (L136–139). Its intuitive explanation is derived from the widely known notion of meta-category in fine-grained data analysis (Related Work, L63–65), so we did not elaborate further before introducing it. Thanks for the reminder. We will consider adding more intuitive explanations before presenting the formal bounds and provide additional background to better introduce key concepts in the theoretical section.
>
> **Response to Questions**
>
> **1.** As mentioned in Limitations and Future Work (Sec. A.2), our experiments follow the standard FG-FSCIL setting, where all classes come from the same domain, and thus do not yet consider cross-domain scenarios. However, datasets like *mini*ImageNet and CIFAR100 already involve varying granularity levels and some degree of domain diversity. Thanks to the two-level feature representation learning, continual adaptation of the feature extractor to novel classes, and continual regularization through class similarity relationships, our method already achieves strong performance on these datasets (Figure 6), demonstrating its potential in settings with domain diversity and varying granularity.
>
> To the best of our knowledge, there is no FSCIL method that specifically considers different domains or varying granularity levels. Thus, it is currently not feasible to obtain established datasets, baselines, and experimental setups conducting targeted experiments in the short term. In future work, we will try to establish such settings and extend MEL by introducing multiple meta-category vectors to better capture domain-specific or differently-grained semantics, and model their interactions to support such scenarios.
>
> **2.** We present an intuitive numerical example to illustrate the effect of Eq. (9) and different optimization strategies.
>
> >Assume the meta-category vector $\mathbf{m} \in \mathbb{R}^2$ is initialized as $\mathbf{m} =$ [1.0, 1.0]. A new session provides three feature vectors $\mathbf{f}_{\text{high}}^i$: [1.2, 1.1], [0.9, 1.0], [4.0, -3.0], with a new class proportion of 0.1. These vectors include two samples that are semantically aligned with the meta-category, and one relatively distinct outlier. The updated meta-category vectors $\mathbf{m}'$, computed using different optimization strategies, are as follows:
> >- Avg: [1.78, 0.03]
> >- Avg + Sim: [1.62, 0.22]
> >- Avg + Weight: [1.10, 0.87]
> >- Ours: [1.06, 0.93]
>
> Compared to other strategies, our method effectively avoids large parameter shifts that may result from few-shot samples, especially mitigating the negative impact of the outlier ([4.0, -3.0]), while incorporating new useful semantic knowledge ([1.2, 1.1], [0.9, 1.0]) from the new session. This is achieved through a *stable update step* via class proportion-based weight $\alpha$ and a *robust direction* via attention weights $\hat{a}_i$. Moreover, these parameters can be adaptively computed based on the data, without the need for manually preset hyperparameters or backpropagation (BP), and thereby improving both practicality and efficiency.
>
>
> We further organize the experimental analysis (Sec. 4.4.1, Figure 4) into the following theoretical analysis.
>
> >From Eq. (9), our meta-category update rule is
> $$\mathbf{m}'=(1-\alpha)\mathbf{m}+\alpha{\textstyle \sum_{i=1}^{m_t}}(\hat{a}_ {i}\mathbf{f}_ {\text{high}}^{i}), $$
> the update magnitude can be bounded as
> $$ \|\|\mathbf{m}' - \mathbf{m} \|\|  = \alpha \left\| \left\| \sum_{i=1}^{m_t} \hat{a}_ i \mathbf{f}_ {\text{high}}^i - \mathbf{m} \right\|\right\|.$$
> **1)** The *update step* is directly controlled by $\alpha = \frac{|\mathcal{C}^{t'}|}{\sum_{t=0}^{t'} |\mathcal{C}^t|}$. As $\alpha$ decreases over incremental sessions, the update becomes increasingly conservative, automatically preventing large deviations in later sessions with limited training data.
> >
> >**2)** $\hat{a}_ i = \frac{\exp(\mathbf{m}^\top \mathbf{f}_ {\text{high}}^i / \sqrt{C_h})}{\sum_{j=1}^{m_t} \exp(\mathbf{m}^\top \mathbf{f}_{\text{high}}^j / \sqrt{C_h})}$.
> These weights assign higher importance to feature vectors more aligned with the current meta vector $\mathbf{m}$, effectively reducing the influence of outliers and guiding the update in a reliable *direction*.
>
> **3.** Fine-grained categories usually belong to the same general meta-category (e.g., all being bird species) (L17–18), which inherently determines that these categories share certain common characteristics. Due to rebuttal format restrictions, we cannot include visualizations, but we offer an illustrative example here. In CUB200, species like the Black-throated Blue Warbler and the Cerulean Warbler have nearly identical wing patterns and beak shapes, so learning to recognize such traits in one class can facilitate representation learning in the other. This latent synergy can be understood as positive knowledge transfer across sub-categories.
>
> **4.** While unidirectional KL divergence enables knowledge transfer from one branch to the other, it introduces asymmetry by treating one branch as the teacher and the other as the student. In contrast, Eq. (14) adopts bidirectional KL to enable mutual regularization and refinement between the two branches, better aligning their sub-category relationship distributions. This helps eliminate redundant information unrelated to category similarity relationship, thereby simplifying classification and reducing the generalization bound by lowering hypothesis space complexity (L203-210, L227-230).
> Empirical results also show that using bidirectional KL divergence results in better performance.
>
> |    | low $\to$ high | high $\to$ low |  bidirectional |
> |-------|:-------:|:------:|:------:|
> | $\text{Acc}_T$ | 43.35 | 43.41 |  44.06 |
> | $\text{Acc}_{Avg}$ | 51.86 | 51.85  |  52.38  |

---

### Official Review · Reviewer_KNQ9 · 2025-07-03

**Clarity:** 4
**Significance:** 3
**Originality:** 3
**Rating:** 5
**Confidence:** 5

**Summary:**

This paper proposes an interesting method, the Meta-Environment Learner (MEL), designed for fine-grained few-shot class-incremental learning. The approach theoretically formulates the problem, deriving a generalization error bound that motivates their design. MEL incorporates two key processes: evolving fine-grained feature extraction across incremental sessions and regularizing hypothesis space complexity. The method incrementally refines a meta-category representation and sub-category relationships. Experiments on four fine-grained benchmarks (CUB200, Stanford Dogs, Stanford Cars, and FGVC Aircraft) demonstrate that MEL achieves superior performance compared to prior state-of-the-art methods. The paper also includes ablation studies and analysis supporting the effectiveness of each component.

**Questions:**

1. The authors only used ResNet-12 as the backbone. Have the authors evaluated MEL with other architectures (e.g., ResNet-18, ResNet-50, or transformers)? Demonstrating that the improvements by the proposed method hold across backbones would strengthen the claim of general applicability.

2. Even though the authors provided the experiments for the effect of hyperparameters, I just want to see how sensitive MEL's performance to the choice of other hyperparameters is.

3. The authors mentioned in the Limitations that the method assumes all classes come from the same domain. Can the authors elaborate on how MEL could be adapted to cross-domain or more heterogeneous settings?

**Ethical Concerns:**

["NO or VERY MINOR ethics concerns only"]

**Final Justification:**

The author provided detailed response with the additional experiments. I keep my accept rating.

**Limitations:**

The authors explicitly discussed limitations in Appendix.

**Paper Formatting Concerns:**

No formatting issues were found.

**Quality:**

3

**Strengths And Weaknesses:**

Strengths
+ The paper tackles an important problem—how to incrementally learn fine-grained categories with few samples, which is highly relevant to real-world applications. And, the authors addressed the motivation and importance of the problem very clearly.
+ The theoretical generalization bound and the concept of progressively evolving a meta-category representation offer fresh perspectives.
+ The comparison experiments were conducted with the recent SoTA methods, and the experimental results demonstrate improvements over strong SoTA methods.
+ The paper is clearly written, with good illustrations and ablation studies explaining design choices.

Weaknesses
- The method is only validated with a ResNet-12 backbone; it would be helpful to see whether it generalizes to other architectures.
- While the method is theoretically motivated, the implementation introduces multiple components whose interactions may be complex to reimplement without open-source code.

---

> ### Author Rebuttal · Authors · 2025-07-30
>
> We sincerely appreciate the constructive feedback. Detailed responses to the weaknesses and questions are as follows.
>
> **Response to Weaknesses**
>
> **1.** Please refer to the first point in the Response to Questions for details.
>
> **2.** To avoid potential reimplementation difficulties, we will release the code if the paper is accepted.
>
> **Response to Questions**
>
> **1.** Thanks for the suggestion. We follow the existing FG-FSCIL setting by using ResNet12 as the backbone. The evaluation results of our method on other architectures are presented below. Our method consistently outperforms the baseline and recent approaches (e.g., PFR, ADBS) across different backbone networks.
>
> |CUB200|  ResNet-18 $\text{Acc}_T$ | ResNet-18 $\text{Acc}_{Avg}$ | ResNet-50 $\text{Acc}_T$ | ResNet-50 $\text{Acc}_{Avg}$ | ViT-B/16 $\text{Acc}_T$ | ViT-B/16 $\text{Acc}_{Avg}$ |
> |-------|:-------:|:------:|:------:|:------:|:------:|:------:|
> | Baseline| 50.17   | 60.05  | 56.79   | 66.39   | 74.89    | 77.90   |
> | PFR   | 57.54  | 65.64   | 58.04   | 67.06   |   78.98  |  81.79  |
> | ADBS  | 60.14  | 67.75   | 61.27   | 68.59   |  79.46   |  82.41   |
> | Ours  | 62.84   | 69.20   | 63.30   | 70.97  |  81.70  |  82.47  |
>
> **2.** The experimental results on the effect of other hyperparameters, the number of attention masks $N$ and the temperature hyperparameter $\tau$, are presented below.
>
> |$N$ |  1  |  2  |  3  |  4  |  5 |  6  |
> |-------|:-------:|:------:|:------:|:------:|:------:|:------:|
> | $\text{Acc}_T$  |   41.77  |  42.94  |  43.15 |  44.06  |  43.55  |  43.32 |
> | $\text{Acc}_{Avg}$|  50.64  |  51.70  |  51.59 |  52.38  |  52.04  |  52.20 |
>
> |$\tau$   |    8    |   12   |   16  |  20  |  24  |
> |-------|:-------:|:------:|:------:|:------:|:------:|
> | $\text{Acc}_T$  |  41.14  |  43.70  |  44.06  |  43.64  | 43.53 |
> | $\text{Acc}_{Avg}$|  51.14  |  52.72  |  52.38  |  52.28  | 51.94 |
>
> MEL's performance remains relatively stable across a range of values for these hyperparameters. When $N \ge 4$, increasing $N$ yields no additional performance gain. Additionally, similar to other FSCIL methods, our MEL achieves better performance when $\tau=16$. These results are consistent with the principles behind our chosen hyperparameter settings (L462-464).
>
> **3.** Our method follows the standard FG-FSCIL setting, where all classes are from the same domain. However, since MEL learns feature representations at two levels, allows continual adaptation of the feature extractor to novel classes, and leverages class similarity relationships to continually regularize learning, it is in principle more adaptable to heterogeneous or cross-domain settings than existing methods. The strong performance of our method on datasets such as *mini*ImageNet and CIFAR100 (Figure 6), which already involve a certain degree of domain diversity, demonstrates its potential to cross-domain settings.
>
> Recently, cross-domain Class Incremental Learning (CIL) [a] has emerged as a novel research direction. However, to the best of our knowledge, there is no existing cross-domain FSCIL method. Given the significance of this direction, in future work, we will establish such a setting and extend MEL by introducing multiple meta-category vectors to better capture domain-specific semantics and modeling their interactions, thereby adapting to cross-domain or more heterogeneous settings.
>
> [a] *Multi-Granularity Class Prototype Topology Distillation for Class-Incremental Source-Free Unsupervised Domain Adaptation. CVPR 2025*

---

> > ### Comment · Area_Chair_53dy · 2025-08-03
> > **Author-reviewer  Discussion period**
> >
> > Dear Reviewers,
> >
> > The NeurIPS 2025 author-reviewer discussion will be closed on August 6, 11:59 pm AoE. Please read the responses, respond to them in the discussion, and discuss points of disagreement.
> >
> > Best,
> > Your AC

---

> ### Comment · Reviewer_KNQ9 · 2025-08-04
>
> I would like to thank the authors for the detailed response with the additional experimental results. I will keep my positive rating. Thanks.

---

### Official Review · Reviewer_5ht6 · 2025-07-03

**Clarity:** 2
**Significance:** 2
**Originality:** 3
**Rating:** 4
**Confidence:** 4

**Summary:**

The paper focus on few-shot class-incremental learning specifically for fine-grained datasets (FG-FSCIL). The authors exploit the fact that sub-categories in fine-grained datasets has smaller inter-class variations compared to conventional datasets. The paper provides a theoretical formulation of FSCIL with generalization bounds within fine-grained meta-category environment. The authors propose a Meta-Environment Learner for FG-FSCIL which evolves the feature extractor in new tasks and also performs regularization. The proposed method outperforms FSCIL baselines across several fine-grained datasets.

**Questions:**

See weaknesses

**Ethical Concerns:**

["NO or VERY MINOR ethics concerns only"]

**Final Justification:**

The authors response addressed my concerns. I raised my initial rating.

**Limitations:**

yes

**Quality:**

2

**Strengths And Weaknesses:**

Strengths -
1. The paper is well-written and easy to understand.
2. The paper provides good theoretical background for FSCIL generalization bounds.

Weaknesses-
1. The impact and scope of the work is very limited since the authors propose the method explicitly only for fine-grained datasets unlike the standard practice of general FSCIL. Only a single paper [15] follows this setting while most FSCIL methods are proposed for general purpose datasets as well as fine-grained datasets.
2. The authors should present some analysis for the claim that sub-categories in fine-grained datasets has smaller inter-class variations and improve the motivation.
3. Other than [15], a recent work [1] also focus more on fine-grained FSCIL using pre-trained weights and propose a very simple and effective statistics calibration method. The authors should include discussions and comparison with [1] in the paper.
4. The method involves several hyper-parameters and the authors do not discuss how these values are decided.

---

> ### Author Rebuttal · Authors · 2025-07-30
>
> We sincerely appreciate the constructive feedback. Detailed responses to the weaknesses and questions are as follows.
>
> **1.** First, while our method is primarily designed for fine-grained datasets, it also demonstrates strong performance on conventional FSCIL benchmarks, as shown in the experiments in Sec. 4.4.3, underscoring the generality of our method.
>
> Second, Fine-grained classification has long been established as a widely studied research direction, and Fine-Grained Few-Shot Class-Incremental Learning (FG-FSCIL) further presents its own unique and significant value.
>
> **1)** From an application perspective, fine-grained data is prevalent in real-world tasks. Compared to conventional datasets, fine-grained datasets often suffer from challenges such as difficulty in data collection and annotation, as well as the continuous emergence of new classes. These challenges naturally align with the FSCIL paradigm (L17-27). Therefore, developing FSCIL methods for fine-grained data holds substantial practical value.
>
> **2)** From a research perspective, the smaller inter-class variations in fine-grained data increase the task difficulty, but also provide an opportunity to enhance synergy across sub-categories, thereby improving FSCIL performance (L32-38). However, this potential is overlooked by existing methods. Exploring how to leverage data characteristics to solve problems not only helps better address the inherent challenges of fine-grained data, but also contributes to the further development of research in the FSCIL field.
>
> As a relatively novel task, FG-FSCIL remains underexplored, and existing works are limited. Nevertheless, given its practical and research value, we believe it will gain broad community attention. And our work, as the first to model fine-grained inter-class synergy from the meta-environment perspective, will have a profound impact. The other reviewers also recognized the importance of the problem we tackle, the important insights and potential impact of our method in promoting progress in the FSCIL field, as well as its ability to address genuine limitations in existing FG-FSCIL methods.
>
> **2.** Thanks for the suggestion. We would like to clarify that the claim *``sub-categories in fine-grained datasets have smaller inter-class variations”* is not specific to our method, but rather a widely acknowledged premise in fine-grained research (L63-66), and thus we did not provide additional analysis on it. As defined in an authoritative survey paper (Fine-Grained Image Analysis With Deep Learning: A Survey. TPAMI 2022): *``the fine-grained nature of the problem is challenging because of the small inter-class variations caused by highly similar sub-categories, and the large intra-class variations in poses, scales and rotations. It is as such the opposite of generic image analysis (i.e., the small intra-class variations and the large inter-class variations), and what makes FGIA a unique and challenging problem.”* We will include further clarification and analysis before this claim in the Introduction.
>
> **3.** Reference [1] in our paper remains a classic FSCIL method, which is relatively older and does not specifically focus on fine-grained FSCIL. Although [1] proposes a clever tree-based statistical approach, it still freezes the feature extractor during incremental sessions and therefore does not exploit the potential synergy among fine-grained sub-categories in feature representation. Under the same experimental settings, its performance on CUB200 is $\text{Acc}\_T$=12.15 and $\text{Acc}_{Avg}$=17.18, which is still clearly inferior to that of our proposed method. We will include a discussion of [1] and add its results to the comparison experiments in the paper.
>
> **4.** Hyperparameter values are decided based on previous works and experiments. More details have been given in supplementary material A.3 (L462-464), and their effect has been analyzed in Sec. 4.4.2. Given the importance of this aspect, we will move the explanation of hyperparameter selection into the main paper.

---

> > ### Comment · Reviewer_5ht6 · 2025-08-01
> >
> > Thanks for the response. I would like to clarify that in the 3rd point of weaknesses, I wanted to refer to [x] but mistakenly did not put the reference. Apologies for the mistake.
> >
> > [x] is one of the few recent works which focus more on fine-grained FSCIL using pre-trained weights and propose a very simple and effective statistics calibration method. Since very few related works in fine-grained FSCIL are discussed in the paper, I believe the authors should include discussions and comparison with [x] in the paper.
> >
> > [x] "Calibrating higher-order statistics for few-shot class-incremental learning with pre-trained vision transformers." Proceedings of the IEEE/CVF Conference on Computer Vision and Pattern Recognition. 2024.

---

> > > ### Author Response · Authors · 2025-08-04
> > >
> > > Thanks for the reference you provided.
> > >
> > > Motivation and Implementation:
> > >
> > > **(1)** Reference [x] focuses on obtaining good higher-order statistics (e.g., covariance matrix) from few-shot data using pre-trained ViT models, and proposes an effective calibration method specifically designed for higher-order statistics-based classification approaches. Although it includes experiments on fine-grained datasets, it remains a FSCIL method without specific discussion or methodological design for the characteristics of fine-grained data.
> > > **(2)** In contrast, we are the first to exploit the potential synergy among sub-categories and, guided by theoretical analysis of FSCIL, design a novel Meta-Environment Learner that simultaneously evolves fine-grained feature extraction and regularizes the hypothesis space complexity.
> > >
> > > Performance:
> > >
> > > **(1)** We tried to adapt the code of [x] to our experimental settings for comparison, but it failed with runtime errors when computing higher-order statistics; and despite further efforts, it could only achieve substantially lower performance. This may be because the design and implementation of [x] rely on pre-trained ViT weights and higher-order statistics-based classification, which are uncommon in FSCIL methods,  and did not consider compatibility with other experimental settings.
> > > **(2)** Thus, we re-implemented our method under the same setting as [x]. The table below shows the average accuracy after the last task reported in [x] across all datasets used in their paper. It can be observed that, based on the same pretrained ViT-B/16 with a ViT adaptor as in [x], our method still achieves a consistent performance advantage on both fine-grained and conventional benchmarks, even though our method is not specifically designed for pretrained ViT weights.
> > >
> > > |   |  CUB200 | FGVC-Aircraft  | Stanford Cars  | CIFAR100
> > > |-------|:-------:|:------:|:------:|:------:|
> > > |[x]   |  83.72  |   40.32   |    65.30    |    85.68
> > > |Ours  |    84.16   |  45.21    |     65.92    |    86.68
> > >
> > > We will include discussions and comparison with [x] in the paper.

---

> > > > ### Comment · Reviewer_5ht6 · 2025-08-06
> > > >
> > > > Thank you for the response. Most of my concerns are addressed.
> > > >
> > > > I believe the comparison with [x] enhances the contributions of the paper showing improved performance using pre-trained ViTs which are being explored in more recent works on FSCIL. The discussions are a valuable addition to the paper. I will update my score based on the above discussion.

---

### Decision · Program_Chairs · 2025-09-17

**Decision:**

Accept (poster)

**Comment:**

This paper addresses the fine-grained setting of few-shot class-incremental learning. Reviewers found it easy to read, tackled important problems, and provided a good theoretical analysis.

During the review process, the paper received some initial negative scores (Reviewer "5ht6"). As the rebuttal process progressed, all reviewers' concerns were addressed.

The paper ultimately received full consensus in the final rating, with two borderline accepts and two accepts. The meta-reviewer agreed that the paper contributes to the field of few-shot class incremental learning and agreed with the reviewers' recommendation for acceptance. The AC urged the authors to include the additional comparative experiments mentioned in the rebuttal and a discussion with recent work in the camera-ready version.